# Assessment of Low-Dose rhBMP-2 and Vacuum Plasma Treatments on Titanium Implants for Osseointegration and Bone Regeneration

**DOI:** 10.3390/ma18153582

**Published:** 2025-07-30

**Authors:** Won-Tak Cho, Soon Chul Heo, Hyung Joon Kim, Seong Soo Kang, Se Eun Kim, Jong-Ho Lee, Gang-Ho Bae, Jung-Bo Huh

**Affiliations:** 1Department of Prosthodontics, Dental Research Institute, Dental and Life Sciences Institute, Education and Research Team for Life Science on Dentistry, School of Dentistry, Pusan National University, Yangsan 50612, Republic of Korea; joonetak@hanmail.net (W.-T.C.); addlee85@naver.com (J.-H.L.); qorkdgh97@naver.com (G.-H.B.); 2Department of Oral Physiology, Periodontal Diseases Signaling Network Research Center, Dental and Life Science Institute, School of Dentistry, Pusan National University, Yangsan 50612, Republic of Korea; snchlheo@gmail.com (S.C.H.); hjoonkim@pusan.ac.kr (H.J.K.); 3Department of Veterinary Surgery, College of Veterinary Medicine and BK21 Plus Project Team, Chonnam National University, Gwangju 61469, Republic of Korea; vetkang@jnu.ac.kr (S.S.K.); ksevet@jnu.ac.kr (S.E.K.)

**Keywords:** titanium implant, rhBMP-2, vacuum plasma, osseointegration, bone regeneration

## Abstract

This study evaluated the effects of low-dose recombinant human bone morphogenetic protein-2 (rhBMP-2) coating in combination with vacuum plasma treatment on titanium implants, aiming to enhance osseointegration and bone regeneration while minimizing the adverse effects associated with high-dose rhBMP-2. In vitro analyses demonstrated that plasma treatment increased surface energy, promoting cell adhesion and proliferation. Additionally, it facilitated sustained rhBMP-2 release by enhancing protein binding to the implant surface. In vivo experiments using the four-beagle mandibular defect model were conducted with the following four groups: un-treated implants, rhBMP-2–coated implants, plasma-treated implants, and implants treated with both rhBMP-2 and plasma. Micro-computed tomography (micro-CT) and medical CT analyses revealed a significantly greater volume of newly formed bone in the combined treatment group (*p* < 0.05). Histological evaluation further confirmed superior outcomes in the combined group, showing significantly higher bone-to-implant contact (BIC), new bone area (NBA), and inter-thread bone density (ITBD) compared to the other groups (*p* < 0.05). These findings indicate that vacuum plasma treatment enhances the biological efficacy of low-dose rhBMP-2, representing a promising strategy to improve implant integration in compromised conditions. Further studies are warranted to determine the optimal clinical dosage.

## 1. Introduction

Dental implants are widely used to restore oral functions such as mastication, speech, and aesthetics in edentulous or partially edentulous patients [1,2]. Titanium is the material of choice due to its high strength, durability, and excellent biocompatibility [2,3,4]. To enhance osseointegration, various implant designs have been developed, incorporating modifications in shape, size, surface texture, and thread geometry [5,6,7]. Among these, surface treatments such as sandblasted, large-grit, and acid-etched (SLA) techniques have demonstrated improved bone-to-implant integration by facilitating osteoblast activity during bone formation [8,9,10]. SLA-treated implants have shown over 90% long-term survival rates in clinical studies [11,12,13,14,15,16]. However, success rates are significantly reduced in patients with systemic conditions such as metabolic disorders or poor bone quality [17,18]. Particularly in diabetic patients, impaired neovascularization and delayed bone remodeling contribute to higher implant failure rates [18]. As a result, biochemical surface modifications have emerged as a promising approach to enhance osseointegration and improve implant outcomes under compromised conditions [19,20,21,22].

Biochemical surface modification using bioactive agents—such as proteins, enzymes, and peptides—has been extensively studied to mimic the properties of natural bone tissue and enhance osseointegration [19,20,21]. Various agents, including extracellular matrix proteins, RGD (Arg-Gly-Asp) peptides, and growth factors, have been applied to improve the biological performance of biomaterials by regulating cellular behavior and tissue responses [22,23,24]. Among these, bone morphogenetic proteins (BMPs), part of the transforming growth factor-beta (TGF-β) superfamily, have demonstrated strong osteoinductive potential by promoting mesenchymal stem cell differentiation and bone matrix synthesis [25,26]. Specifically, BMP-2, -4, -6, -7, and -9 have shown efficacy in bone regeneration, particularly when delivered via collagen carriers in orthopedic applications [26]. Recombinant human BMP-2 (rhBMP-2) has been successfully used in dental procedures such as sinus augmentation and ridge preservation [27,28,29]. However, clinical use of rhBMP-2 has raised safety concerns, including ectopic bone formation, inflammation, adipogenesis, osteoclastic activity, and potential tumorigenesis [30,31]. A critical issue in dental applications is the lack of consensus on the minimum effective dose required for bone regeneration [31]. High-dose or uncontrolled release of rhBMP-2 has frequently resulted in adverse effects. This highlights the necessity of developing delivery systems that allow for the sustained release of low-dose rhBMP-2 directly from implant surfaces [32,33,34,35]. Although a clear definition of “low-dose” rhBMP-2 remains elusive, concentrations in the range of 0.05–0.5 mg/mL have been employed in several studies as low-dose formulations [36,37]. However, statistically significant effects on bone regeneration were not observed at these concentrations [36]. Consequently, it has been proposed by some researchers that sustained release of rhBMP-2 may enhance its osteoinductive efficacy [38,39].

Various mechanical and chemical strategies have been investigated to immobilize rhBMP-2 on implant surfaces [2,40,41]. Mechanical methods, such as surface roughening, aim to increase surface area for protein loading but often require higher amounts of bioactive agents [2]. Chemical methods involve covalent bonding using cross-linkers like APTES or EDC-NHS; however, residual chemicals may induce cytotoxic or immunogenic responses [42,43]. To overcome these limitations, recent research has focused on radiation- or plasma-based techniques to enable low-dose bioactive agent immobilization without harmful byproducts [7,44,45]. Photoelectric methods, including electron beam and gamma irradiation, have been used for both sterilization and cross-linking, and have shown efficacy in loading bioactive agents onto dental implants in preclinical models [7,45]. Nevertheless, their clinical application is hindered by high cost, limited accessibility, and the risk of surface degradation upon air exposure [7]. Non-thermal atmospheric plasma (NTAPP) has emerged as a promising alternative in dentistry, effectively enhancing the surface energy and hydrophilicity of titanium implants [44]. However, NTAPP requires a continuous gas supply and exhibits limitations in treatment uniformity due to its pen- or needle-type plasma delivery system [44,46]. Recently, tabletop vacuum plasma systems have been introduced for the disinfection of dental instruments and reactivation of contaminated titanium surfaces [46,47]. While ultraviolet (UV) treatment has been used for similar purposes, it is limited by prolonged processing times and transient surface effects [48]. In contrast, vacuum plasma generates high-energy radicals in a low-pressure environment, enhancing surface energy without the need for carrier gases such as argon or nitrogen [49,50,51,52,53,54]. It also introduces functional groups (e.g., –COOH, –OH, –NH_2_) onto the titanium surface, which promote fibronectin adhesion and upregulate osteogenic markers such as osteopontin and osteocalcin [51,55].

Based on these properties, a method was developed to uniformly coat low-dose rhBMP-2 onto titanium implants via vacuum plasma treatment. This technique increases surface energy, creates a bioactive environment, and avoids residual contaminants. The present study aimed to evaluate the biological effects of this combined approach—low-dose rhBMP-2 coating with vacuum plasma treatment—by comparing it with implants treated with rhBMP-2 alone or plasma treatment alone in terms of osseointegration and bone regeneration.

## 2. Materials and Methods

### 2.1. In Vitro Study

#### 2.1.1. Classification of Investigational Groups

Titanium SLA disks (10 mm in diameter, 4 mm in height, PNUADD, Busan, Republic of Korea) were used for the in vitro experiments. These disks were fabricated using the same titanium material, machining process, and sandblasted and acid-etched (SLA) surface treatment protocol as the SLA implants used in the in vivo study. Both the disks and implants were produced by the same manufacturer under identical processing conditions.

To evaluate the effects of low-dose rhBMP-2 and vacuum plasma-treated SLA implants on cellular performance, the specimens were divided into the following groups:Group NS: Non-treated SLA disks;Group BS: SLA disks coated with low-dose rhBMP-2;Group PS: Vacuum plasma-treated SLA disks;Group PB: SLA disks treated with vacuum plasma irradiation followed by low-dose rhBMP-2 coating.

Group NS was an un-treated SLA disk, and group BS had 20 μg of 0.05 mg/mL rhBMP-2 applied to the surface of the SLA disk. Group PS was plasma-irradiated from a plasma generator for 10 s on the SLA disk. For group PB, 20 μg of 0.05 mg/mL rhBMP-2 was applied to the SLA disk after irradiating plasma from a plasma generator for 10 s. All SLA disks were washed with PBS buffer 8 h after applying rhBMP-2 and dried in a dry oven for 24 h.

#### 2.1.2. Scanning Electron Microscope (SEM) Analysis

For morphological surface analysis, FE-SEM (Tescan MIRA, Brno, Czech Republic) imaging was performed for all four groups. Prior to imaging, the SLA disks from each group were coated with a conductive layer using an ion sputter coater to enhance surface conductivity. SEM was operated at an accelerating voltage of 10 keV to obtain surface images of the disks.

#### 2.1.3. X-Ray Photoelectron Spectroscopy (XPS) Analysis

The surface chemical composition of the plasma-treated SLA disk coated with rhBMP-2 was investigated through X-ray photoelectron spectroscopy (XPS; NEXSA, Thermo Fisher Scientific, Pittsburgh, PA, USA), utilizing a monochromatic Al Kα radiation source (1486.6 eV) and a spot diameter of 400 μm. Spectral data were interpreted using Vision 1.5 software (Version 2.2.6, Kratos Analytical Ltd., Manchester, UK).

#### 2.1.4. Cell Culture and Conditions

Human dental pulp stem cells (hDPSCs; ScienCell Research Laboratories, Carlsbad, CA, USA) were maintained in α-minimum essential medium (α-MEM) containing 10% fetal bovine serum (FBS) and penicillin–streptomycin. When the cells reached approximately 90% confluency, subculturing was performed using trypsin–EDTA.

#### 2.1.5. Alkaline Phosphatase (ALP) Activity Assay

To assess alkaline phosphatase (ALP) activity, SLA disks with or without plasma treatment were positioned in 48-well culture plates. Each well received 500 μL of phosphate-buffered saline (PBS) containing different concentrations of rhBMP-2. Following incubation at 37 °C for designated time intervals, cells were seeded onto each disk at a density of 20,000 cells per well. The cells were then cultured for 5 additional days, with the culture medium replaced every 2 days. ALP activity was quantified using the 1-Step PNPP substrate solution (Thermo Fisher Scientific, Waltham, MA, USA), following the manufacturer’s protocol. Absorbance was read at 405 nm.

#### 2.1.6. Real-Time Quantitative Reverse Transcription Polymerase Chain Reaction (qRT-PCR)

Total RNA was extracted from cells cultured on SLA disks using the RNeasy Mini Kit (Qiagen), following the manufacturer’s protocol. Two micrograms of the isolated RNA were reverse-transcribed into cDNA using SuperScript II (Invitrogen, Waltham, MA, USA) under standard conditions. For quantitative PCR (qPCR), 40 ng of the synthesized cDNA was combined with SYBR Green PCR Master Mix (Applied Biosystems, Foster City, CA, USA) and subjected to 40 amplification cycles using an AB7500 system (Applied Biosystems). All reactions were performed in triplicate, and β-actin was used as the internal control for normalization. Gene expression levels were calculated using the 2^−ΔΔCt method. The sequences of the primers used are listed below:

**Runx2:** 5′-CCCTGAACTCTGCACCAAGT-3′, 5′-TGGAGTGGATGGATGGGGAT-3′;

**OCN:** 5′-GCAATAAGGTAGTGAACAGACTCC-3′, 5′-GTTTGTAGGCGGTCTTCAAGC-3′;

**ALP:** 5′-TGACCTTCTCTCCTCCATCC-3′, 5′-CTTCCTGGGAGTCTCATCCT-3′;

**β-actin:** 5′-TCTGGCACCACACCTTCTAC-3′, 5′-TACGACCAGAGGCATACAGG-3′.

#### 2.1.7. Cumulative Release

Un-treated and plasma-treated SLA disks were positioned in 48-well culture plates, and each well received 500 μL of PBS containing 2 μg/mL of rhBMP-2. Following a 2 h incubation at 37 °C, the disks were transferred into fresh 48-well plates containing 1 mL of DPBS and maintained under gentle agitation at 50 rpm at 37 °C. Supernatants were collected at predetermined intervals (1, 3, 6, and 12 h; and 1, 3, 7, 14, and 21 days), and the DPBS was refreshed after each collection. The harvested solutions were stored at −20 °C until analysis. The concentration of rhBMP-2 released over time was quantified using a commercially available ELISA kit (R&D Systems, Minneapolis, MN, USA) in accordance with the manufacturer’s guidelines. Absorbance was determined at 450 nm using an Opsys MR microplate reader.

### 2.2. In Vivo Study

#### 2.2.1. Classification of Investigational Groups

The twenty-four dental implants (ADDplant ON, PNUADD, Busan, Republic of Korea) used in this study had a diameter of 3.5 mm and a length of 8.5 mm. Although the group nomenclature was consistent with that in Section 2.1.1, SLA implants were used as the experimental material in this in vivo study. A commercially available vacuum plasma generator (ACTILINK™ Reborn, Plasmapp, Seoul, Republic of Korea) was used to irradiate implants in Groups PS and PB. Plasma processing consisted of the following three stages: vacuum, irradiation, and purification. The vacuum pump maintained the pressure in the chamber at approximately 5 torr, and a dielectric barrier discharge was generated using a power supply at a frequency of 100 kHz and a voltage of 3 kV.

For Groups BS and PB, 20 μL of 0.05 mg/mL rhBMP-2 was applied to the Ti implant surface.

#### 2.2.2. Experimental Animals

This study was approved by the Institutional Animal Care and Use Committee (IACUC) of the Chonnam National University Biomaterial R&BD Center (BMC-IACUC-2023-08(01)). Four healthy male beagles (12 months old, 10 ± 2 kg; DooYeol Biotech, Seoul, Republic of Korea) were used in this study. Animals with a medical history indicating systemic disease, eating disorders, pregnancy, abnormal body weight, or pre-existing dental pathology were excluded from the study. There were no animals, experimental units, or data points excluded from the analysis. Humane endpoints were established based on external clinical signs, including anorexia and sustained weight loss exceeding 20% of normal body weight, inability to walk (increased time spent in lateral recumbency), severe organ or systemic symptoms, and moribund state indicating imminent death. Animals exhibiting any of these signs were to be humanely euthanized to minimize suffering.

#### 2.2.3. Surgical Procedure

Prior to both surgical procedures, general anesthesia was induced in all beagles using a combination of medetomidine (Tomidin, Provet, Istanbul, Turkey) at a dosage of 10 μg/kg and tiletamine–zolazepam (Zoletil 50^®^, Virbac Laboratories, Carros, France) at 5 mg/kg, followed by maintenance with isoflurane inhalation anesthesia (Sevoflurane, Hana Pharm Co., Seoul, Republic of Korea). For pain management during anesthesia, tramadol (Maritrol, Cheil Pharmaceutical, Yongin-si, Republic of Korea) at 2 mg/kg and carprofen (Rimadyl Inj., Zoetis, Parsippany, NJ, USA) at 2.2 mg/kg were delivered intravenously. Local anesthesia was provided at the surgical site through nerve block injection of 0.4 mL bupivacaine (Bupivacaine HCl 0.5% Inj., Myungmoon Pharm Co., Seoul, Republic of Korea). To minimize infection risk, cefazolin sodium (Cefazolin^®^, Chongkundang Pharm Co., Seoul, Republic of Korea) was administered at a dose of 20 mg/kg.

Following full-mouth scaling, the premolars (P1–P4) and first molars (M1) of the mandible were extracted. The extraction sites were sutured with 4-0 Vicryl (Mersilk, Ethicon Co., Livingston, UK). After the initial surgery, an 8-week healing period was allowed to ensure adequate recovery from tooth extractions.

The surgical protocol is depicted in Figure 1. Implant placement was conducted in a second surgical session 8 weeks after the extractions, using the same anesthesia and post-operative management as previously described. Occlusal cylindrical bone defects (5.5 mm in diameter, 3 mm in depth from the alveolar crest) were prepared using cylindrical trephine and carbide burs under constant saline irrigation.

A total of 24 implants from four different experimental groups were randomly distributed into the prepared defects using a random number generator. To assess peri-implant bone regeneration, three threads of each implant were intentionally left exposed from the base of the defect. Implant stability was evaluated by attaching a SmartPeg to each fixture and measuring the implant stability quotient (ISQ) using an Osstell device (W&H Dentalwerk, Bürmoos GmbH, Bürmoos, Austria). To reduce bias, implant placements were carried out in randomized order, and all beagles were maintained under standardized environmental conditions, including temperature, humidity, and light/dark cycles.

Wound closure was completed using 4-0 Vicryl sutures (Mersilk, Ethicon Co., Livingston, UK). Eight weeks post-implantation, the animals were humanely euthanized under general anesthesia, after which the mandibles were retrieved and ISQ values were re-assessed.

#### 2.2.4. Volumetric Analysis Using Medical Computed Tomography (Medical CT)

Volumetric analysis using medical CT was performed to compare bone volumes in regions of interest (ROIs) based on CT data acquired immediately after the second surgery and at the time of sacrifice (Figure 2). Medical CT scans were obtained using a Philips Incisive CT system (WCT-667-140, Philips Healthcare (Suzhou) Co., Ltd., Suzhou, China) under the following conditions: a tube voltage of 120 kV and a rotation speed of 0.35 s.

#### 2.2.5. Volumetric Analysis Using Micro-Computed Tomography (μ-CT)

Micro-CT (μ-CT) analysis was additionally performed to obtain more precise volumetric measurements of newly formed bone around the implants. Mandibles were scanned using a μ-CT system (SkyScan 1273, Bruker-CT Co., Kartuizersweg 3B, 2550 Kontich, Belgium) at a current of 55 μA and a voltage of 80 kV to acquire μ-CT data for the regions of interest (ROIs) (Figure 3). A pixel resolution of 24.15 μm was used to determine new bone volume (NBV) in the peri-implant defects. The reconstructed μ-CT images were processed using NRecon reconstruction software (version 1.7.0.4, Bruker-CT Co., Kontich, Belgium).

#### 2.2.6. Histological Analysis

The harvested specimens were initially fixed in 10% neutral-buffered formalin at ambient temperature and subsequently dehydrated through a graded ethanol series (70%, 80%, 90%, and 100%) (Duksan Pure Chemical Co., Ltd., Gyeonggi-do, Republic of Korea). Following dehydration, the samples were infiltrated with Technovit 7200 resin (Heraeus Kulzer, Hanau, Germany) for one week. After full infiltration, the specimens were placed in an embedding mold and polymerized using a UV curing system (EXAKT 520, Heraeus Kulzer, Norderstedt, Germany).

The cured blocks were sectioned through the center of the implants into 400 μm-thick slices using a diamond band saw (EXAKT 300CP, Exakt Apparatebau, Norderstedt, Germany). These slices were then ground and polished to a final thickness of approximately 30 μm using a precision grinding system (EXAKT 400CS, Exakt Apparatebau, Norderstedt, Germany). Finalized sections were mounted on microscope slides and stained with Goldner’s trichrome (GT). Digital images of stained sections were captured using a slide scanner and evaluated using CaseViewer software (version 2.1, 3DHISTECH Ltd., Budapest, Hungary).

Histomorphometric parameters, including bone-to-implant contact (BIC), inter-thread bone density (ITBD), and new bone area (NBA), were measured by a trained investigator using ImageJ software (Version 1.54m, US NIH, Bethesda, MD, USA). The region of interest (ROI) was defined as the area covering the top three exposed threads and a 1 mm perimeter around each implant, as illustrated in Figure 4.(1)$$\mathrm{NBA}\left(\%\right)=\frac{\mathrm{New bone area}\;({\mathrm{mm}}^{2})}{\mathrm{Total ROI area}\;({\mathrm{mm}}^{2})}\times 100$$(2)$$\mathrm{BIC}\left(\%\right)=\frac{\mathrm{Length of the new bone to implant contact}\;({\mathrm{mm}}^{2})}{\mathrm{Total ROI length of implant}\;({\mathrm{mm}}^{2})}\times 100$$(3)$$\mathrm{ITBD}\left(\%\right)=\frac{\mathrm{New bone area of inter thread}\;({\mathrm{mm}}^{2})}{\mathrm{Total area of inter thread}\;({\mathrm{mm}}^{2})}\times 100$$

### 2.3. Statistical Analysis

For in vitro experiments, comparisons between two groups were conducted using Student’s *t*-test, while differences among three or more groups were analyzed via one-way or two-way analysis of variance (ANOVA), followed by Bonferroni post hoc correction. A *p*-value of less than 0.05 was considered statistically significant. Data are presented as mean values ± standard deviation (SD), and all experiments were independently repeated at least three times.

For the in vivo study, data are expressed as means ± SD, and statistical analyses were performed using SPSS version 27 (SPSS Inc., Chicago, IL, USA). Normality tests confirmed that NBA, ITBD, BIC, and new bone volume (NBV) measured by both medical CT and μ-CT were normally distributed. Therefore, one-way ANOVA was used to evaluate differences between groups, followed by Fisher’s least significant difference (LSD) post hoc test. Statistical significance was accepted at *p* < 0.05.

## 3. Results

### 3.1. In Vitro Study

#### 3.1.1. Morphological Findings

Surfaces of the four groups were observed at ×25, ×1000, and ×5000 magnifications. Rough SLA surfaces were noted in Groups NS and PS, which were not coated with rhBMP-2 (Figure 5). Group BS, coated with rhBMP-2, exhibited lumps of rhBMP-2 attached to the SLA surface. In Group PB, plasma treatment increased surface energy, and rhBMP-2 was observed to penetrate the spaces between the rough surfaces.

#### 3.1.2. X-Ray Photoelectron Spectroscopy (XPS) Findings

Representative XPS spectra of the four groups treated with plasma and/or rhBMP-2 on titanium disks are shown in Figure 6. The surface chemical compositions of the four groups are summarized in Table 1. Compared to the control Group NS, Group PS treated with plasma alone showed lower carbon and increased oxygen levels. Additionally, Groups BS and PB, treated with rhBMP-2, exhibited relatively high carbon and nitrogen contents and low oxygen levels. In Groups BS and PB, coated with rhBMP-2, Ti2p signals were almost undetectable, indicating that rhBMP-2 covers the titanium surface.

#### 3.1.3. Alkaline Phosphatase (ALP) Activity

Analysis of ALP activity at various treatment times and rhBMP-2 concentrations showed that disks treated with 50 μg/mL rhBMP-2 for 120 min exhibited higher ALP activity compared to treatment for 30 min and 360 min (Figure 7). Furthermore, when evaluating ALP activity at the 120 min treatment time, rhBMP-2 concentrations of 5 and 50 μg/mL resulted in significantly higher ALP activity.

#### 3.1.4. Real-Time Quantitative Reverse Transcription Polymerase Chain Reaction (qRT-PCR)

The osteogenic differentiation of C2C12 cells in response to rhBMP-2 released from the implants was further characterized by analyzing osteogenic-specific genes using qRT-PCR (Figure 8). Among the tested genes, OCN and ALP—but not Runx2—were significantly upregulated by rhBMP-2 coating following plasma treatment. These findings indicate that plasma treatment enhances the efficacy of rhBMP-2 coating on the implants, thereby promoting osteogenic differentiation.

#### 3.1.5. Cumulative Release

To confirm whether plasma treatment effectively enhanced the immobilization of rhBMP-2 on the implant surface, we evaluated the cumulative release of rhBMP-2 in the supernatant using ELISA. As shown in Figure 9, the release profile of rhBMP-2 exhibited the following typical two-phase pattern: a relatively rapid initial release within the first 12 h, followed by a sustained and slower release over an extended period. Over the 21-day release test, the concentration of rhBMP-2 in the supernatant from control implants (54.03 ± 4.54 ng/mL) was higher than that from plasma-treated implants (42.39 ± 3.46 ng/mL). These results suggest that plasma pretreatment increased rhBMP-2 immobilization on the implants, thereby reducing its release into the supernatant.

### 3.2. In Vivo Findings

#### 3.2.1. Clinical Findings

All beagles survived healthily without infection or inflammation after both the first and second surgical procedures.

#### 3.2.2. ISQ Findings

The ISQ results are shown in Table 2. ISQ values increased on average at each measurement time point, but no significant differences were found among the groups. Regarding changes, Group PB in the buccal area and Group PS in the lingual area showed relatively large changes before and after implantation; however, these differences were not statistically significant due to high standard deviations.

#### 3.2.3. Medical CT Findings

Medical CT images have relatively low resolution and were therefore used to analyze changes in bone volume from immediately after implantation to 8 weeks post-operation, rather than to measure the precise volume of newly formed bone (Figure 10).

The results of NBV (%) are presented in Table 3. No statistically significant differences in NBV were observed among the groups when measured using medical CT at each time point. However, the change in NBV from baseline to 8 weeks was highest in Group PB compared to the other groups. Although one-way ANOVA analysis yielded a *p*-value of 0.063, which was not statistically significant, the post hoc Fisher’s LSD test revealed that Group PB exhibited a significantly greater increase in NBV than the other groups.

#### 3.2.4. Micro-CT Findings

Micro-CT imaging confirmed the formation of new bone within the peri-implant bone defects in all groups (Figure 11). Notably, in Group PB, newly formed bone extended beyond the original defect margin, reaching the implant platform. In contrast, buccal exposure of the implant threads was observed in Group NS and BS up to the second thread, and up to the first thread in Group PS. Importantly, no buccal thread exposure was observed in Group PB. Regardless of group, vertical bone loss on the buccal side of the pre-existing bone was commonly observed.

The results of NBV (%) measured using μ-CT are presented in Table 4. The lowest NBV was observed in Group NS (52.87%), while the highest was found in Group PB (75.00%). Groups BS and PS showed comparable NBV values of 64.51% and 65.56%, respectively. Statistical analysis revealed that all experimental groups demonstrated significantly higher NBV compared to Group NS (*p* < 0.05). However, no significant differences were found between Groups BS and PS, or between Groups PS and PB.

#### 3.2.5. Histological Findings

The histological analysis results are presented in Figure 12. All groups showed normal tissue healing without any signs of abnormal inflammatory cell infiltration or pathologic changes. New bone regeneration was observed within the created defects, even in the absence of bone graft material. However, buccal bone loss was commonly observed across many specimens. Notably, in Group PB, newly formed bone completely covered the previously exposed three implant threads, whereas one or two threads remained exposed in the other groups.

#### 3.2.6. Histometric Findings

The histometric analysis results are summarized in Table 5. The new bone area (NBA) values were highest in Group PB (72.75% ± SD), followed by Groups BS (70.46%), PS, and NS, in descending order. Although Group PB had the highest NBA, the difference was not statistically significant when compared to Group BS. However, Group NS showed a significantly lower NBA than the other groups.

For bone-to-implant contact (BIC), Group PB again demonstrated the highest value (83.91% ± SD), with statistically significant differences compared to other groups. Groups BS and PS showed similar BIC values (68.81% and 70.98%, respectively), with no significant difference between them.

In terms of inter-thread bone density (ITBD), Group PB recorded the highest average value (81.32% ± SD), which was significantly greater than the other groups. Groups BS (63.47%) and PS (64.81%) showed similar outcomes without significant differences, while Group NS had the lowest ITBD at 42.86%, showing a statistically significant difference from the others.

## 4. Discussion

Many studies have been conducted to promote osseointegration and bone regeneration without the use of bone grafts or membranes by coating titanium implant surfaces with bioactive substances such as extracellular matrix proteins, RGD peptides, and growth factors [22,23,24]. Among these, rhBMP-2 has been shown to promote the differentiation of mesenchymal stem cells into osteoblasts, which are crucial for new bone formation [25,26]. Consequently, rhBMP-2 has demonstrated strong osteoinductive potential in clinical studies, showing vertical bone regeneration even in compromised host conditions [56]. Wikesjö et al. [56] reported vertical bone augmentation and improved bone quality at all tested concentrations (0.75, 1.5, and 3.0 mg/mL) when 59 titanium implants coated with rhBMP-2 were placed in alveolar bone and evaluated at 4 and 8 weeks. Additionally, previous clinical studies using 0.5 mg/mL rhBMP-2-coated implants also confirmed bone regeneration effects [57]. However, there are currently no established guidelines in dentistry for determining the minimum effective dose of rhBMP-2 based on patient age or weight [32]. In orthopedic procedures such as spinal fusion for large osseous defects, a concentration of 1.5 mg/mL of rhBMP-2 has typically been employed [26,30]. While this dosage has proven effective in promoting osteoinduction, it has also been associated with a higher risk of adverse effects, including edema, excessive bone formation, inflammation, and cyst formation [30,31]. The rationale for using such high concentrations has been attributed to the short biological half-life of rhBMP-2 [32]. Due to its rapid absorption and clearance in vivo, the protein often fails to persist long enough to exert its osteoinductive or regenerative effects on osteogenic cells [33,34,35]. In response to these complications, previous studies have evaluated lower concentrations of rhBMP-2 (0.05, 0.1, and 0.2 mg/mL) for their effects on osteoinduction and osseointegration [36]. However, these studies reported that such low concentrations did not result in statistically significant effects osteoinduction and osseointegration [36]. Therefore, in the present study the lowest concentration previously reported (0.05 mg/mL) was selected for evaluation. The dosage was selected based on significant ALP activity observed in vitro.

Although rhBMP-2 is well-known for its osteoinductive potential, several studies have reported inconsistent outcomes regarding its efficacy in bone augmentation or regeneration [33,58]. These discrepancies may be attributed to the premature release of rhBMP-2 from carrier systems, which limits its bioavailability at the target site [22,40]. To address this limitation, sustained-release strategies have been explored, including polymer coatings such as collagen [2,40,41]. However, these polymer-based carriers are susceptible to mechanical damage during implantation and may provoke adverse immune responses [59]. In the present study, vacuum plasma treatment was employed to facilitate uniform coating of low-dose rhBMP-2 onto SLA-treated titanium implants. This method effectively increased surface energy and improved hydrophilicity without leaving harmful residues. Cumulative release testing confirmed stable bonding between rhBMP-2 and the SLA surface, supporting a controlled and prolonged release profile. SEM analysis further revealed rhBMP-2 retention within the micro-porous structure of the SLA surface, corroborating the improved adhesion and surface integration.

SLA surface treatment itself is clinically established for enhancing implant stability by creating a micro-rough topography [8,60]. However, hydrocarbon contamination during storage can reduce surface energy and compromise osseointegration and bioactive agent loading [32,47,57]. Plasma treatment has the added benefit of removing such contaminants, thereby restoring surface reactivity and facilitating more efficient biomolecule attachment [61,62]. Compared to UV or radiation methods, vacuum plasma offers practical advantages, including compact equipment and gas-free operation, making it more clinically applicable [50,51]. In vivo, the PB group (plasma + BMP-2 + SLA) demonstrated significantly greater bone-to-implant contact (BIC), inter-thread bone density (ITBD), and new bone area (NBA) compared to other groups. These histometric results suggest that the combination of plasma-treated surfaces with low-dose rhBMP-2 provides a synergistic effect on osseointegration and bone regeneration. Micro-CT analysis confirmed a higher new bone volume in the PB group (75.04%) than in the NS and BS groups, and longitudinal medical CT imaging showed the greatest volume increase (30.11%) in the PB group at 8 weeks post-operatively. Interestingly, no significant differences in implant stability were observed among the groups, despite differences in BIC and bone volume. This suggests that commonly used clinical stability measures such as RFA may not fully capture subtle enhancements from surface biochemical modifications alone. These findings are consistent with previous studies reporting weak correlations between bone–implant contact and RFA values [63].

Within the limitations of this preliminary study—including small sample size, short follow-up period, and the use of a single rhBMP-2 concentration—the results indicate that vacuum plasma-assisted rhBMP-2 coating enhances both bone regeneration and osseointegration. Further research is warranted to determine the optimal rhBMP-2 dosage and to assess long-term outcomes and clinical applicability.

## 5. Conclusions

Within the limitations of this study, vacuum plasma treatment effectively facilitated the sustained release of low-dose rhBMP-2 coated on SLA-treated titanium implants. Implants treated with both vacuum plasma and low-dose rhBMP-2 exhibited significantly enhanced bone regeneration and osseointegration in a beagle mandibular defect model. These findings suggest that plasma-assisted coating may be a promising strategy for improving implant performance. However, further studies are warranted to establish the optimal concentration and long-term clinical efficacy of rhBMP-2.

## Figures and Tables

**Figure 1 materials-18-03582-f001:**
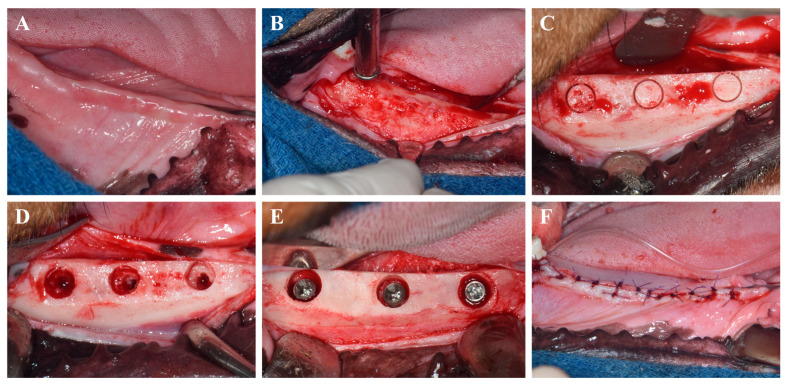
Surgical procedures for implant placement. (**A**) Intraoral photograph before surgery. (**B**) Flap elevation and alveolar bone flattening. (**C**) Marking of implant sites for defect creation. (**D**) Creation of occlusal cylindrical defects. (**E**) Allocation of implants according to experimental groups. (**F**) Suturing of the gingiva.

**Figure 2 materials-18-03582-f002:**
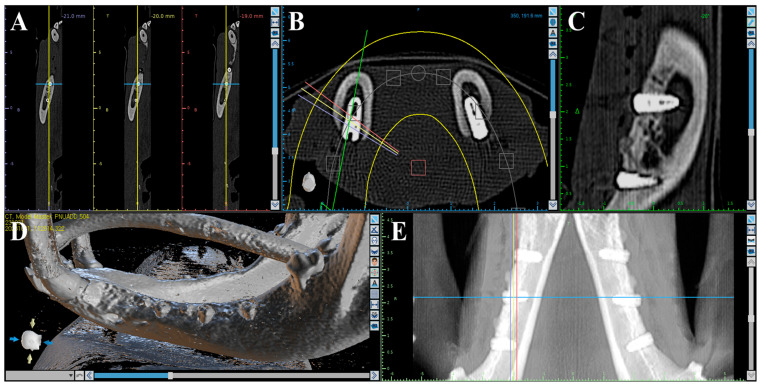
Medical CT images used for volumetric analysis. (**A**) Axial view. (**B**) Coronal view. (**C**) Sagittal view. (**D**) 3D rendering. (**E**) Panoramic view.

**Figure 3 materials-18-03582-f003:**
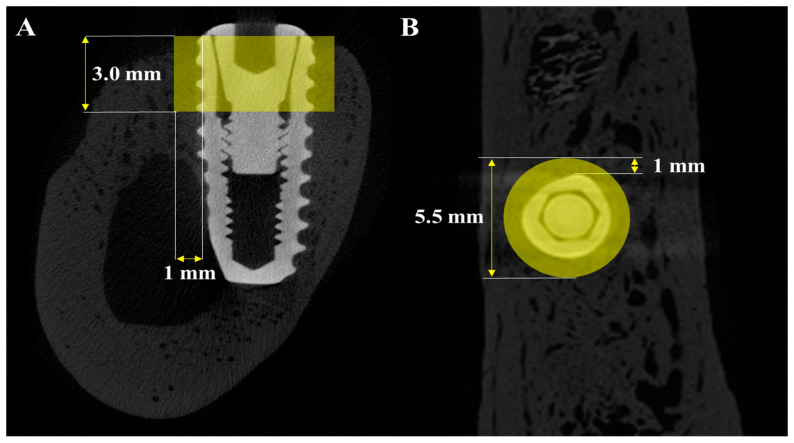
μ-CT images used for volumetric analysis of peri-implant bone. (**A**) Sagittal section showing the vertical and horizontal dimensions of the region of interest (ROI). (**B**) Axial section indicating the circular ROI around the implant.

**Figure 4 materials-18-03582-f004:**
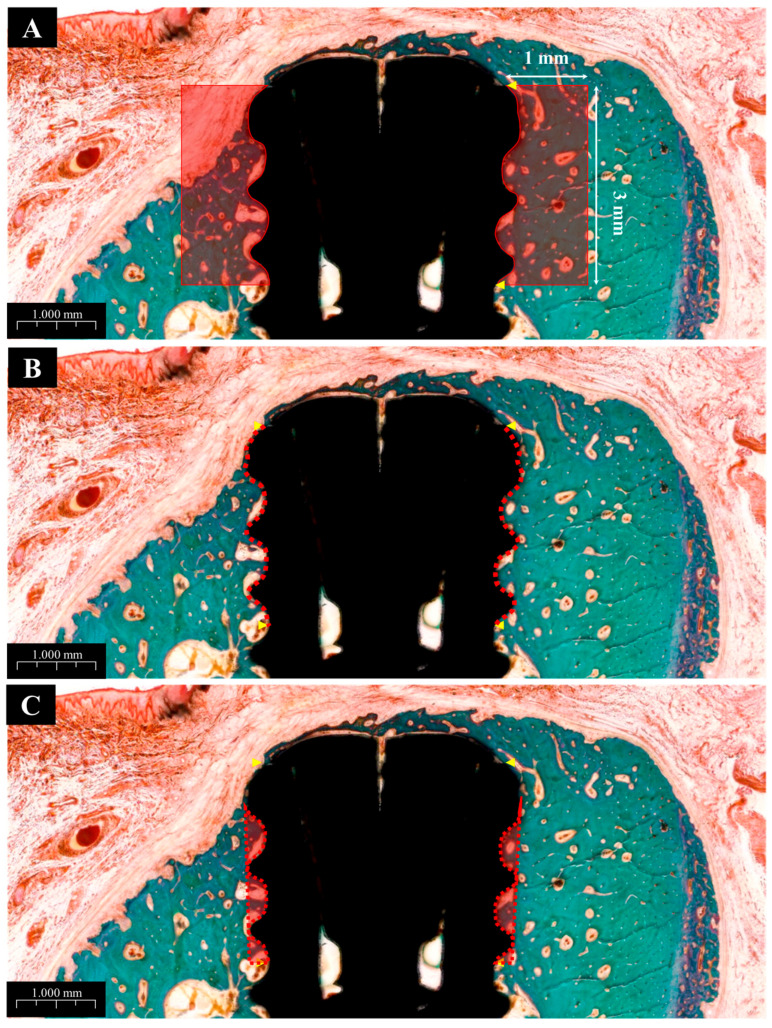
Histological images used for histometric analysis. (**A**) Defined ROI area used to calculate new bone area (NBA), bone-to-implant contact (BIC), and inter-thread bone density (ITBD). (**B**) Measurement of BIC. (**C**) Measurement of ITBD.

**Figure 5 materials-18-03582-f005:**
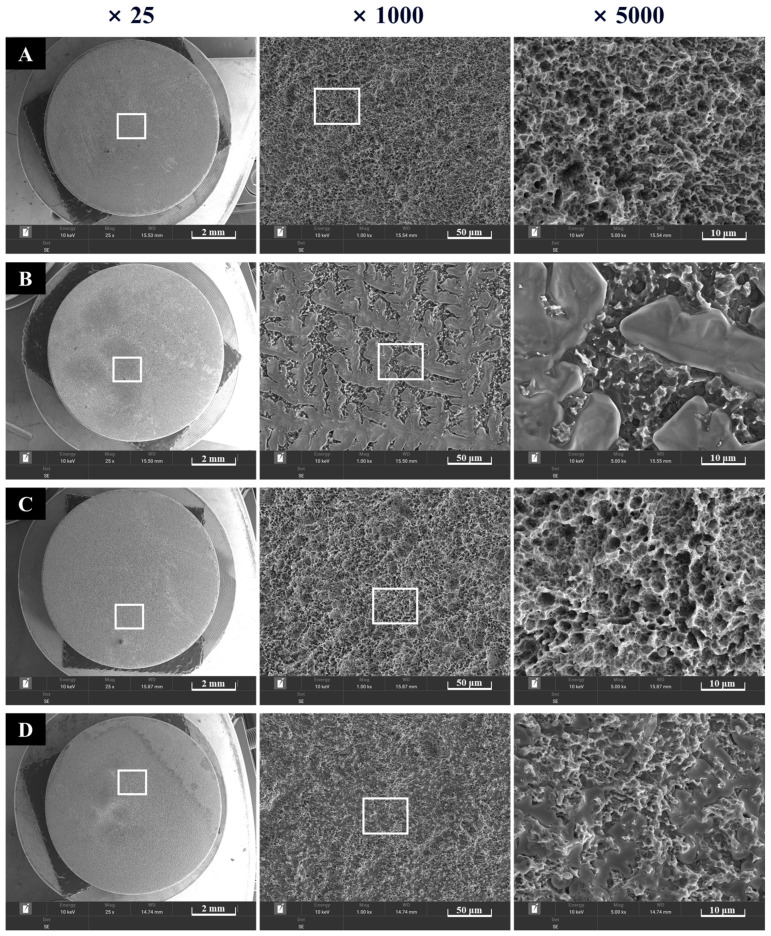
Morphological images of implant surfaces at ×5000 (left) and ×15,000 (right) magnification. (**A**) Group NS; non-treated SLA disks. (**B**) Group BS; SLA disks coated with low-dose rhBMP-2. (**C**) Group PS; vacuum plasma-treated SLA disks. (**D**) Group PB; SLA disks treated with vacuum plasma irradiation followed by low-dose rhBMP-2 coating.

**Figure 6 materials-18-03582-f006:**
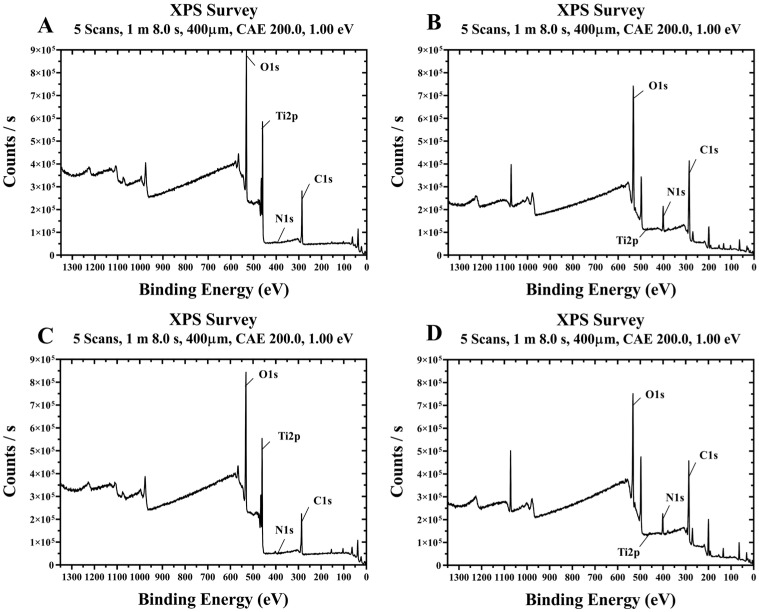
Survey XPS spectra of titanium disks from each group. (**A**) Group NS; non-treated SLA disks. (**B**) Group BS; SLA disks coated with low-dose rhBMP-2. (**C**) Group PS; vacuum plasma-treated SLA disks. (**D**) Group PB; SLA disks treated with both low-dose rhBMP-2 and vacuum plasma.

**Figure 7 materials-18-03582-f007:**
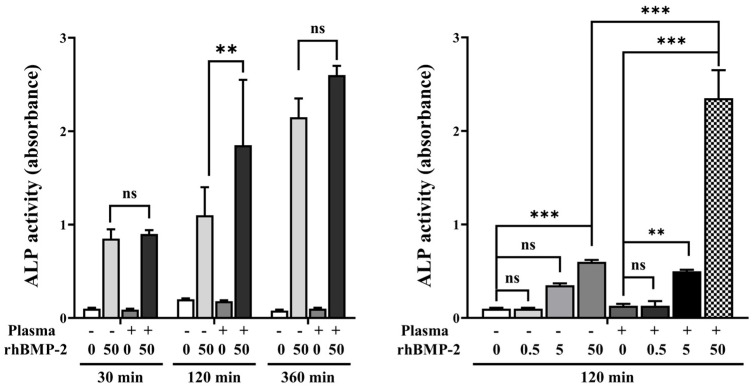
ALP activity according to treatment time and rhBMP-2 concentration. ns, not significance. ** Indicates statistical significance (*p* < 0.01). *** Indicates statistical significance (*p* < 0.001).

**Figure 8 materials-18-03582-f008:**
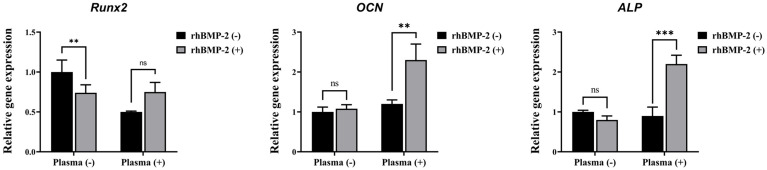
Real-time PCR analysis of C2C12 cells. ns, not significance. ** Indicates statistical significance (*p* < 0.01). *** Indicates statistical significance (*p* < 0.001).

**Figure 9 materials-18-03582-f009:**
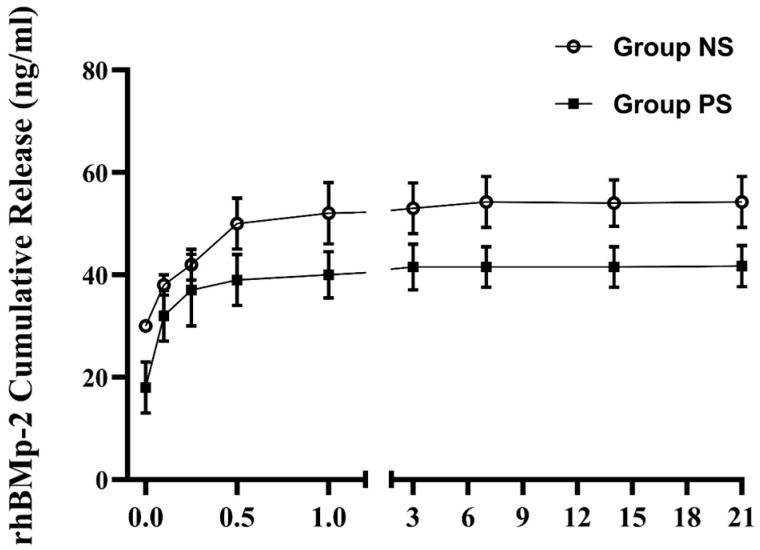
Cumulative release kinetics of rhBMP-2 from the SLA surface.

**Figure 10 materials-18-03582-f010:**
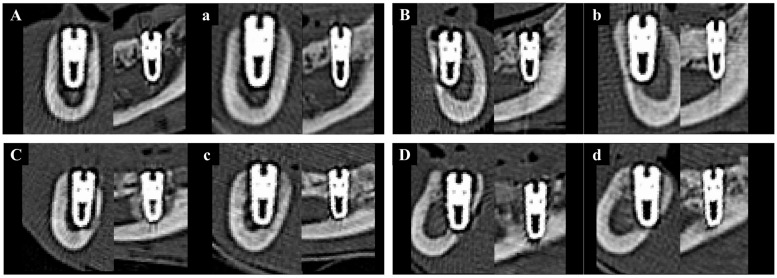
Representative sectional medical CT images of each group at baseline (0 weeks) and 8 weeks post-implantation. Left column: coronal views; right column: sagittal views. (**A**) Group NS at 0 wk. (**a**) Group NS at 8 wk. (**B**) Group BS at 0 wk. (**b**) Group BS at 8 wk. (**C**) Group PS at 0 wk. (**c**) Group PS at 8 wk. (**D**) Group PB at 0 wk. (**d**) Group PB at 8 wk.

**Figure 11 materials-18-03582-f011:**
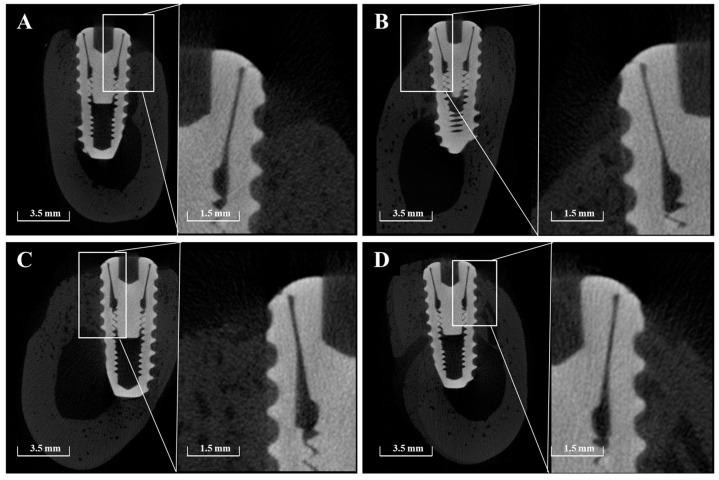
Representative sectional micro-CT images of each group at 8 weeks post-implantation. (**A**) Group NS; non-treated SLA implants. (**B**) Group BS; SLA implants coated with low-dose rhBMP-2. (**C**) Group PS; vacuum plasma-treated SLA implants. (**D**) Group PB; SLA disks treated with vacuum plasma irradiation followed by low-dose rhBMP-2 coating.

**Figure 12 materials-18-03582-f012:**
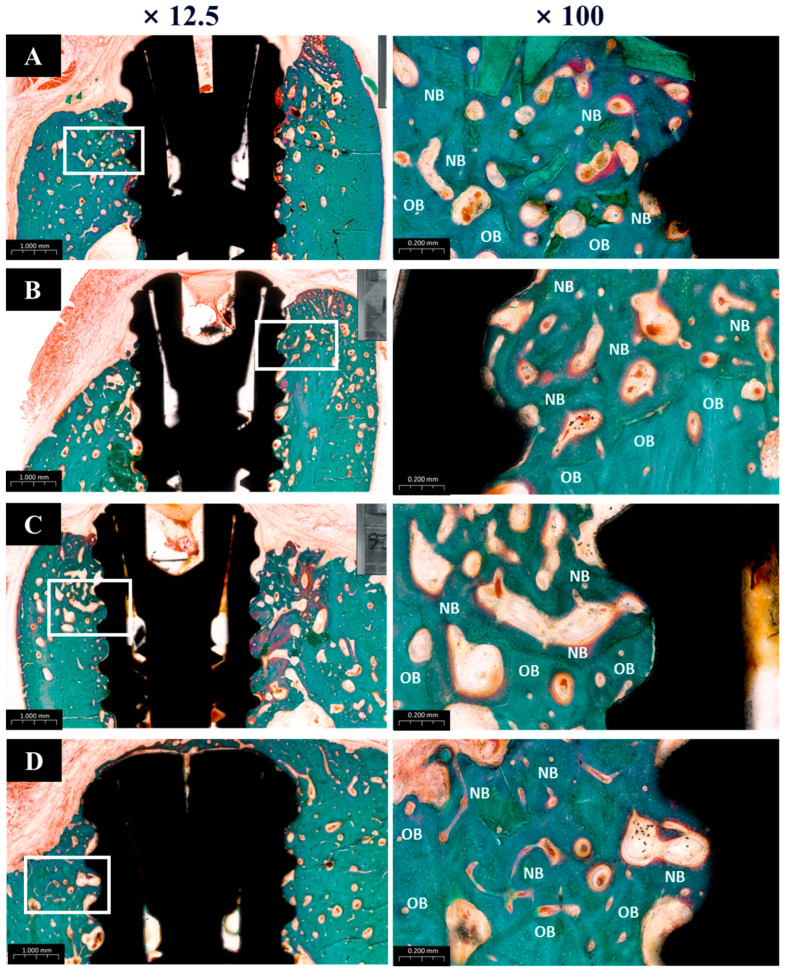
Goldner’s trichrome (GT) stain histological sections of each groups. (**A**) Group NS; non-treated SLA implants. (**B**) Group BS; SLA implants coated with low-dose rhBMP-2. (**C**) Group PS; vacuum plasma-treated SLA implants. (**D**) Group PB; SLA disks treated with vacuum plasma irradiation followed by low-dose rhBMP-2 coating. NB; new bone. OB; old bone.

**Table 1 materials-18-03582-t001:** Surface chemical composition of titanium disks for each group.

Compositions (at%)	Elements			
Groups	C1s	N1s	O1s	Ti2p
NS	34.07	1.04	43.15	21.73
BS	56.83	8.78	34.1	0.3
PS	31.23	1.36	46.2	21.21
PB	59.67	7.22	32.55	0.56

Group NS; non-treated SLA disks. Group BS; SLA disks coated with low-dose rhBMP-2. Group PS; vacuum plasma-treated SLA disks. Group PB; SLA disks treated with vacuum plasma irradiation followed by low-dose rhBMP-2 coating.

**Table 2 materials-18-03582-t002:** Implant stability quotient (ISQ) values and changes at buccal and lingual sites over 8 weeks.

Contents	Sites	Groups	Mean ± SD		
			0 wk	8 wk	Change (%)
ISQ	Buccal	NS	61.00 ± 6.66	65.80 ± 4.07	8.60 ± 7.98
		BS	60.43 ± 8.36	66.00 ± 6.05	10.49 ± 11.91
		PS	58.13 ± 7.90	65.63 ± 12.34	12.45 ± 10.73
		PB	59.10 ± 7.03	67.20 ± 7.67	14.45 ± 13.13
	*p*-Value		*p* > 0.05	*p* > 0.05	*p* > 0.05
	Lingual	NS	62.80 ± 4.71	64.40 ± 5.89	2.48 ± 4.53
		BS	60.57 ± 8.96	66.57 ± 6.61	11.37 ± 12.46
		PS	58.25 ± 7.76	68.25 ± 10.49	17.19 ± 10.12
		PB	59.20 ± 6.97	67.20 ± 7.61	14.32 ± 13.79
	*p*-Value		*p* > 0.05	*p* > 0.05	*p* > 0.05

Implant stability quotient; ISQ. Group NS; non-treated SLA disks. Group BS; SLA disks coated with low-dose rhBMP-2. Group PS; vacuum plasma-treated SLA disks. Group PB; SLA disks treated with vacuum plasma irradiation followed by low-dose rhBMP-2 coating.

**Table 3 materials-18-03582-t003:** New bone volume (NBV, %) and changes from baseline to 8 weeks post-implantation assessed by medical CT.

Contents	Weeks	Groups	Mean ± SD	*p*-Value
NBV (%)	0	NS	32.41 ± 5.20	*p* > 0.05
		BS	39.98 ± 5.43	
		PS	39.29 ± 7.26	
		PB	32.94 ± 10.15	
	8	NS	53.06 ± 12.09	*p* > 0.05
		BS	59.21 ± 8.32	
		PS	55.45 ± 10.55	
		PB	63.06 ± 3.73	
	Changes	NS	20.64 ± 12.36 ^a^	0.063
		BS	19.23 ± 8.21 ^a^	
		PS	16.17 ± 10.56 ^a^	
		PB	30.11 ± 7.22 ^b^	

New bone volume; NBV. Group NS; non-treated SLA disks. Group BS; SLA disks coated with low-dose rhBMP-2. Group PS; vacuum plasma-treated SLA disks. Group PB; SLA disks treated with vacuum plasma irradiation followed by low-dose rhBMP-2 coating. ^a,b^ Different lowercase letters indicate significant differences.

**Table 4 materials-18-03582-t004:** New bone volume (NBV, %) results at 8 weeks post-surgery as assessed by micro-computed tomography (μ-CT).

Contents	Weeks	Groups	Mean ± SD	*p*-Value
NBV (%)	8	NS	52.87 ± 10.78 ^a^	0.002 *
		BS	64.51 ± 8.33 ^b^	
		PS	65.56 ± 6.22 ^b,c^	
		PB	75.04 ± 6.33 ^c,d^	

* Indicates statistical significance (*p* < 0.05). New bone volume; NBV. Group NS; non-treated SLA implants. Group BS; SLA implants coated with low-dose rhBMP-2. Group PS; vacuum plasma-treated SLA implants. Group PB; SLA disks treated with vacuum plasma irradiation followed by low-dose rhBMP-2 coating. ^a–d^ Different lowercase letters indicate significant differences.

**Table 5 materials-18-03582-t005:** Mean values of new bone area (NBA), bone-to-implant contact (BIC), and inter-thread bone density (ITBD) in histometric analysis (mean ± SD, %).

Contents	Groups	Mean ± SD	*p*-Value
NBA (%)	NS	43.79 ± 10.60 ^a^	0.000 *
	BS	70.46 ± 5.17 ^b,c^	
	PS	61.76 ± 6.59 ^b^	
	PB	72.75 ± 9.36 ^c^	
BIC (%)	NS	60.24 ± 3.51 ^a^	0.000 *
	BS	68.81 ± 6.40 ^b,e^	
	PS	70.98 ± 6.63 ^c,e^	
	PB	83.91 ± 3.01 ^d^	
ITBD (%)	NS	42.86 ± 10.44 ^a^	0.000 *
	BS	63.47 ± 6.03 ^b,e^	
	PS	64.81 ± 7.32 ^c,e^	
	PB	81.32 ± 6.62 ^d^	

* Indicates statistical significance (*p* < 0.05). New bone area; NBA. Bone-to-implant contact; BIC. Inter threads bone density; ITBD. Group NS; non-treated SLA implants. Group BS; SLA implants coated with low-dose rhBMP-2. Group PS; vacuum plasma-treated SLA implants. Group PB; SLA disks treated with vacuum plasma irradiation followed by low-dose rhBMP-2 coating. ^a–e^ Different lowercase letters indicate significant differences.

## Data Availability

The original contributions presented in this study are included in the article. Further inquiries can be directed to the corresponding authors.

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
