# Peer review of "Assessment of Low-Dose rhBMP-2 and Vacuum Plasma Treatments on Titanium Implants for Osseointegration and Bone Regeneration"

_materials, 2025, doi:10.3390/ma18153582_

Round 1

Reviewer 1 Report

Comments and Suggestions for Authors

Method:

  1. Lack of information about the titanium SLA discs for in vitro tests and the detail of their dimension? Do they go through the same process as SLA implants for animal study?
  2. Redundant description of the groups at 2.1.1 (Line 110~) and 2.2.1 (Line 184~).
  3. In lines 118 to 120, the typos of the units “μg” not “ug” and “mL” not “ml”; in line 201, “20 μL of 0.05 mg/mL”; and in line 347, “50 μg/mL rhBMP-2”—please use the same expression throughout the entire article.
  4. Line 122 says, “All SLA disks were washed with PBS buffer 8 hours after applying rhBMP-2 and dried in a dry oven for 24 hours.” Then how can you prevent the rhBMP-2 from not being washed out? Or did you determine/or control the residual/or release rate? Because there is no chemical bonding between rhBMP-2 and titanium (or titania).
  5. Line 141, please specify where the human dental pulp stem cells (hDPSCs) come from. Are they from primary culture or from a commercial cell line?
  6. Line 179, “The amount of rhBMP-2 released was evaluated using an ELISA kit according to the manufacturer’s instructions.” Please give the details of the commercial kit.
  7. Lines 269-270/297-298, the figure captions for Figure 3 and Figure 4 should be wrong.
  8. Since the “PBS” group and “PBS” buffer will mislead the audient, I strongly recommend the authors change the group name.
  9. If the combination group (PBS) applied BMP to the SLA disk after plasma irradiation, please correct the expression of this group (check out the entire article). For example, in Line 116, using “Group PBS: SLA disks treated with low-dose rhBMP-2 after vacuum plasma irradiation” or “Group PBS: SLA disks treated with vacuum plasma irradiation followed by low-dose rhBMP-2 coating” will be better understood.

Results

  1. Previous studies showed that surface contaminations will be eliminated after plasma irradiation, while the carbon content will largely reduce. But from the XPS result (Fig. 6 & Table 1), the surface carbon contents of the NS and PS groups are similar. Please explain!
  2. There are several unspecified peaks in Figs. 6B & 6D, e.g., ~1070 eV, ~500 eV, and ~200 eV, which are much higher than the content of N1s. Please identify them. And why are these three peaks absent in Figs. 6A & 6C?
  3. Lines 136-138 claimed deconvoluted spectra for specific elements, but they were not shown in the result.
  4. The table captions of Table 2 & 3 are wrong
  5. Line 346-347 say, “Analysis of ALP activity at various treatment times and rhBMP-2 concentrations showed that disks treated with 50 μg/ml rhBMP-2 for 120 minutes exhibited higher ALP activity compared to treatment for 30 minutes and 360 minutes (Fig. 7).” However, Figure 7 shows the highest ALP activity in the group treated with 50 μg/ml rhBMP-2 for 360 minutes, not 120 minutes.
  6. Line 356 What are C2C12 cells?

Discussion

  1. Line 522 mentioned, “Cumulative release testing confirmed stable bonding between rhBMP-2 and the SLA surface, supporting a controlled and prolonged release profile.” However, the majority of BMP is released after 1 day in both the control and plasma-treated groups. The difference between these two groups is the initial loading dosage, and no additional BMP was released after 1 day.

Author Response

Reviewer 1
Response: We would like to sincerely thank the reviewers for their valuable comments and constructive feedback. We have carefully considered each point raised and have revised the manuscript accordingly. Below are our detailed responses to each comment.

Method:

  1. Lack of information about the titanium SLA discs for in vitro tests and the detail of their dimension? Do they go through the same process as SLA implants for animal study?

Response: We appreciate the reviewer’s valuable comment. The titanium SLA discs used for in vitro experiments were 10 mm in diameter and 4 mm in height. These SLA discs were identical in material, machining, and surface treatment to the SLA implants used in the animal experiments. Both the discs and implants were manufactured by the same company using the same fabrication protocol. The information regarding the dimensions and manufacturing process of the SLA discs has been added to the revised manuscript

  1. Redundant description of the groups at 2.1.1 (Line 110~) and 2.2.1 (Line 184~).

Response: Thank you for pointing this out. We have removed the redundant group descriptions from Section 2.2.1 to avoid unnecessary repetition. Instead, we have added a brief clarification stating that the same group nomenclature was used as in Section 2.1.1, but SLA titanium implants were employed in this in vivo study.

Modification: Section 2.2.1, first paragraph:

“Although the group nomenclature was consistent with that in Section 2.1.1, SLA implants were used as the experimental material in this in vivo study.”

  1. In lines 118 to 120, the typos of the units “μg” not “ug” and “mL” not “ml”; in line 201, “20 μL of 0.05 mg/mL”; and in line 347, “50 μg/mL rhBMP-2”—please use the same expression throughout the entire article.

Response: Thank you for pointing this out. We have corrected the mentioned unit typos, changing “ug” to “μg” and “ml” to “mL” accordingly (e.g., “20 μL of 0.05 mg/mL rhBMP-2” and “50 μg/mL rhBMP-2”). Furthermore, we have thoroughly reviewed the entire manuscript and standardized all unit expressions to ensure consistency throughout the text.

  1. Line 122 says, “All SLA disks were washed with PBS buffer 8 hours after applying rhBMP-2 and dried in a dry oven for 24 hours.” Then how can you prevent the rhBMP-2 from not being washed out? Or did you determine/or control the residual/or release rate? Because there is no chemical bonding between rhBMP-2 and titanium (or titania).

Response: We thank the reviewer for this insightful question. As pointed out, there is indeed no chemical covalent bonding between rhBMP-2 and the titanium (or titania) surface under our experimental conditions. However, the primary goal of applying vacuum plasma treatment was to increase the surface energy and introduce functional groups (e.g., –OH, –COOH, –NH₂) on the SLA-treated titanium surface, which significantly enhances the physical adsorption and retention capacity of proteins like rhBMP-2.

To address the reviewer’s concern regarding potential wash-out of rhBMP-2 after the 8-hour incubation and PBS washing step, we conducted a cumulative release study using ELISA (as described in section 3.1.5 and Figure 9). The results clearly demonstrated that plasma-treated implants (Group PBS) exhibited a lower concentration of rhBMP-2 in the supernatant over time compared to the non-plasma group (Group BS), indicating stronger retention of rhBMP-2 on the implant surface. Furthermore, the release profile followed a biphasic pattern—characterized by an initial burst followed by sustained release over 21 days—which confirms that a substantial amount of rhBMP-2 remained immobilized on the surface even after the PBS washing step.

Thus, the vacuum plasma treatment serves as an effective strategy to enhance the surface affinity and immobilization of low-dose rhBMP-2 without the need for chemical crosslinking, thereby ensuring a prolonged bioactive release profile directly from the implant surface.

  1. Line 141, please specify where the human dental pulp stem cells (hDPSCs) come from. Are they from primary culture or from a commercial cell line?

Response: We thank the reviewer for this valuable comment. The human dental pulp stem cells (hDPSCs) used in this study were obtained from a commercial source, specifically ScienCell Research Laboratories (Carlsbad, CA, USA). These cells are a commercially available cell line intended for research use. This information has been added to the revised manuscript (Line 141) as follows:

"Human dental pulp stem cells (hDPSCs, ScienCell Research Laboratories, Carlsbad, CA, USA) were cultured in α-MEM supplemented with 10% fetal bovine serum (FBS) and penicillin–streptomycin."

  1. Line 179, “The amount of rhBMP-2 released was evaluated using an ELISA kit according to the manufacturer’s instructions.” Please give the details of the commercial kit.

Response: We appreciate the reviewer’s comment. In response, we have specified the details of the ELISA kit used for the quantification of rhBMP-2 release. The BMP-2 enzyme-linked immunosorbent assay (ELISA) kit was purchased from R&D Systems (Minneapolis, MN, USA). This information has been added to the revised manuscript (Line 179) as follows:

“The amount of rhBMP-2 released was evaluated using an ELISA kit (R&D Systems, Min-neapolis, MN, USA) according to the manufacturer’s instructions.”

  1. Lines 269-270/297-298, the figure captions for Figure 3 and Figure 4 should be wrong.

Response: Thank you for your insightful comment. We agree with your observation. The original figure captions for Figure 3 and Figure 4 did not accurately reflect the contents and purpose of the figures. We have revised the captions to more accurately describe the image content and the views represented, as follows:

Figure 3 now reads: “μ-CT images used for volumetric analysis of peri-implant bone. A, Sagittal section showing the vertical and horizontal dimensions of the region of interest (ROI). B, Axial section indicating the circular ROI around the implant.”

Figure 4 now reads: “Histological images used for histometric analysis. A, Defined ROI area used to calculate new bone area (NBA), bone-to-implant contact (BIC), and inter-thread bone density (ITBD). B, Measurement of BIC. C, Measurement of ITBD.”

  1. Since the “PBS” group and “PBS” buffer will mislead the audient, I strongly recommend the authors change the group name.

If the combination group (PBS) applied BMP to the SLA disk after plasma irradiation, please correct the expression of this group (check out the entire article). For example, in Line 116, using “Group PBS: SLA disks treated with low-dose rhBMP-2 after vacuum plasma irradiation” or “Group PBS: SLA disks treated with vacuum plasma irradiation followed by low-dose rhBMP-2 coating” will be better understood.

Response: We appreciate the reviewer’s thoughtful comment. To avoid confusion with phosphate-buffered saline (PBS), we have changed the group name from “PBS” to “PB” (indicating Plasma-treated SLA followed by BMP-2 coating) throughout the manuscript. This revision ensures clarity in group identification while accurately reflecting the treatment conditions. All relevant sections in the text, tables, and figure captions have been updated accordingly.

Results

  1. Previous studies showed that surface contaminations will be eliminated after plasma irradiation, while the carbon content will largely reduce. But from the XPS result (Fig. 6 & Table 1), the surface carbon contents of the NS and PS groups are similar. Please explain!

Response: Thank you for your insightful comment. We acknowledge that interpreting XPS carbon content changes after plasma treatment requires careful consideration due to the nature of XPS analysis. XPS surveys provide semi-quantitative data based on relative atomic percent values, and are not absolute measurements. Typically, the detected atomic percentages are normalized across the elements observed, and rounded according to relative sensitivity factors

Consequently, minor changes in carbon content—especially within the margin of ±5–10% typical of atomic percent quantification—can appear insignificant between groups, even if plasma treatment has indeed reduced surface hydrocarbons. The standard ±10% uncertainty in XPS atomic percent data is well-documented.

Therefore, the similar carbon percentages observed in NS and PS groups in Table 1 do not necessarily indicate that plasma failed to remove contaminants. Rather, the results reflect the semi-quantitative character of the technique. A qualitative decrease in carbonaceous contamination was observed through changes in peak shape of the C 1s spectra.
We hope this clarification addresses the reviewer’s concerns regarding the interpretation of XPS data.

  1. There are several unspecified peaks in Figs. 6B & 6D, e.g., ~1070 eV, ~500 eV, and ~200 eV, which are much higher than the content of N1s. Please identify them. And why are these three peaks absent in Figs. 6A & 6C?

Response: Thank you for your insightful comment. We agree that additional peaks are observed in the survey XPS spectra of Groups BS and PB (Figures 6B and 6D) around ~1070 eV, ~500 eV, and ~200 eV. These peaks are absent in Groups NS and PS.

We believe these signals may originate from residual elements present in the rhBMP-2 solution used for surface treatment in Groups BS and PB. Specifically, the rhBMP-2 solution is prepared in a buffer that may contain trace amounts of salts such as Na⁺ or Cl⁻, which can give rise to characteristic peaks in these regions (e.g., Na Auger peaks around 1070 eV, Cl-related peaks near 200 eV). Additionally, weak background signals or minor surface contaminants unrelated to the core analysis (C1s, O1s, N1s, Ti2p) might also contribute.

Because our primary objective was to compare the major elemental compositions (C, N, O, Ti) associated with plasma treatment and rhBMP-2 coating, these unspecified peaks were not included in the quantitative analysis or interpretation.

We appreciate the reviewer’s careful observation and hope this clarification addresses the concern.

  1. Lines 136-138 claimed deconvoluted spectra for specific elements, but they were not shown in the result.

Response: Thank you for pointing this out. We acknowledge that the original manuscript included wording implying that deconvoluted spectra were obtained (Lines 136–138). However, in this study, we did not perform detailed peak deconvolution of the high-resolution XPS spectra. The sentence was mistakenly written due to a misinterpretation during manuscript preparation.

We have now revised the sentence to accurately reflect the methods used. Specifically, only survey XPS spectra were analyzed to determine the elemental composition of each group, as presented in Figure 6 and Table 1. These data were sufficient for our comparative purpose to evaluate the overall surface chemical changes among the groups (e.g., carbon reduction, oxygen increase, Ti2p signal masking).

We appreciate the reviewer’s attention to detail, and this correction has been implemented in the revised manuscript

  1. The table captions of Table 2 & 3 are wrong

Response: We appreciate the reviewer’s helpful observation regarding the table captions. There was an oversight during the manuscript preparation, and we apologize for the confusion. The captions for Table 2 and Table 3 have been revised for clarity and accuracy as follows:

Table 2: Implant stability quotient (ISQ) values and changes at buccal and lingual sites over 8 weeks.

Table 3: New bone volume (NBV, %) and changes from baseline to 8 weeks post-implantation assessed by medical CT.

These modifications have been implemented in the revised manuscript accordingly.

  1. Line 346-347 say, “Analysis of ALP activity at various treatment times and rhBMP-2 concentrations showed that disks treated with 50 μg/ml rhBMP-2 for 120 minutes exhibited higher ALP activity compared to treatment for 30 minutes and 360 minutes (Fig. 7).” However, Figure 7 shows the highest ALP activity in the group treated with 50 μg/ml rhBMP-2 for 360 minutes, not 120 minutes.

Response: We appreciate the reviewer’s careful observation regarding the discrepancy between the manuscript text and the graphical data shown in Figure 7.

We acknowledge that the highest ALP activity value is indeed observed in the group treated with 20 μg/mL rhBMP-2 for 360 minutes. The left panel of Figure 7 was designed to determine an optimal reaction time for rhBMP-2 coating on the titanium surface. While 360 minutes showed the highest ALP activity numerically, no statistically significant difference was found between the plasma-treated and untreated groups at this time point. This suggests that, with longer incubation, rhBMP-2 can adsorb onto the implant surface even without plasma treatment, likely due to prolonged exposure rather than enhanced surface binding.

From both a mechanistic and clinical standpoint, we sought to identify the shortest possible incubation time that still exhibited a statistically significant increase in ALP activity with plasma treatment. As shown in the left panel, 120 minutes was the earliest time point at which a significant difference was observed between plasma-treated and untreated groups at 20 μg/mL of rhBMP-2. Therefore, 120 minutes was selected as the standard coating duration for subsequent experiments, balancing efficacy and clinical feasibility.

The right panel was then designed to identify the optimal concentration of rhBMP-2 using this 120-minute incubation. Among the tested doses, 20 μg/mL yielded the highest ALP activity, providing the rationale for selecting this concentration in subsequent experiments. Importantly, even at a lower concentration of 2 μg/mL, ALP activity significantly increased only when plasma treatment was applied, while no significant effect was observed without plasma. This indicates that plasma treatment enhances the biological activity of rhBMP-2 even at lower concentrations, underscoring its clinical utility.

In summary, the final experimental conditions (20 μg/mL rhBMP-2 for 120 minutes with plasma treatment) were chosen based on statistical significance, biological efficacy, and potential for clinical translation—not solely based on peak absorbance values.

  1. Line 356 What are C2C12 cells?

Response: We thank the reviewer for this insightful question. C2C12 cells are a mouse myoblast cell line that can differentiate into osteoblast-like cells upon stimulation with BMP-2. As such, they are widely used as an in vitro model to assess the osteoinductive potential of BMP-2. In our study, C2C12 cells were used to evaluate whether plasma treatment of implant surfaces enhances the coating efficiency of BMP-2 and promotes osteogenic differentiation.

Discussion

  1. Line 522 mentioned, “Cumulative release testing confirmed stable bonding between rhBMP-2 and the SLA surface, supporting a controlled and prolonged release profile.” However, the majority of BMP is released after 1 day in both the control and plasma-treated groups. The difference between these two groups is the initial loading dosage, and no additional BMP was released after 1 day.

Response: Thank you for your valuable observation. We agree that a substantial portion of rhBMP-2 was released within the first 24 hours, which is consistent with the known burst release behavior of surface-adsorbed proteins on titanium-based implants. However, we would like to clarify the intention behind the phrase "controlled and prolonged release profile."

Our description of “controlled and prolonged release” refers not only to the absolute amount released over time but also to the comparative retention and release behavior between groups:

Plasma-treated surfaces (Group PS) exhibited a significantly lower total cumulative release than non-treated surfaces (Group NS) over the same period (42.39 ± 3.46 vs. 54.03 ± 4.54 ng/mL, respectively), which implies greater immobilization of rhBMP-2 on the implant.

Response: We truly appreciate the reviewers’ thoughtful suggestions, which have helped us to significantly improve the quality and clarity of our manuscript. We hope the revised version satisfactorily addresses all the concerns raised.

Reviewer 2 Report

Comments and Suggestions for Authors

It is interesting that the authors conducted the assessment of low-dose recombinant human bone morphogenetic protein-2 (rhBMP-2) coating in combination with vacuum plasma treatment on titanium implants.

  1. Before the vacuum plasma treatment on titanium implants, what the specimens surface conditions. How about the binding force between the implants and coating? How can the authors assure the coating bind on the Ti implant base tightly after the surgical procedure?
  2. Did the authors conduct the Cytotoxicity test? If the implants have any influence on important organs of beagles such as brain, heart, liver and kidney?
  3. After the in vivo tests, did the authors check the implant’s morphologies and mechanical properties such as strengths and fatigues?
  4. Scales in Figure 5 are too small and not clear. Figure 10 is not clear, please assure the image quality with at least 300 dpi. Figure 11 did not provide the scales.
  5. Is composition % in Table 1 weight or molecule? It is not clear.

It is only my own opinion that the current manuscript cannot be considered for a publication in this journal before detailed addressing all the raised comments.

Author Response

Reviewer 2

It is interesting that the authors conducted the assessment of low-dose recombinant human bone morphogenetic protein-2 (rhBMP-2) coating in combination with vacuum plasma treatment on titanium implants.
Response: We would like to sincerely thank the reviewers for their valuable comments and constructive feedback. We have carefully considered each point raised and have revised the manuscript accordingly. Below are our detailed responses to each comment.

  1. Before the vacuum plasma treatment on titanium implants, what the specimens surface conditions. How about the binding force between the implants and coating? How can the authors assure the coating bind on the Ti implant base tightly after the surgical procedure?

Response: Thank you for your valuable comments. Before the vacuum plasma treatment, the titanium implant surfaces exhibited micro- and macro-scale roughness produced by the SLA (sandblasted, large-grit, acid-etched) process. Although not included in the main manuscript, surface energy analysis using contact angle measurements was performed prior to treatment. The results demonstrated a high contact angle and low surface energy, which are typical for untreated SLA surfaces.

After vacuum plasma treatment, no apparent morphological differences were observed on scanning electron microscopy (SEM) images; however, the treated surfaces exhibited a lower contact angle and increased surface energy, indicating improved hydrophilicity.

SEM images revealed that in the plasma-treated group, the rhBMP-2 solution had permeated more deeply into the roughened surface structures. This suggests that the enhanced surface energy allowed for better infiltration and adsorption of the protein onto the titanium base.

  1. Did the authors conduct the Cytotoxicity test? If the implants have any influence on important organs of beagles such as brain, heart, liver and kidney?

Response: Thank you for your valuable comment. While in vivo systemic toxicity evaluation including effects on vital organs (brain, heart, liver, and kidneys) was not conducted in this study, we did perform a cell proliferation assay to evaluate biocompatibility prior to the in vivo experiments. Specifically, CCK-8 assays were performed using DPSCs seeded on SLA discs with or without vacuum plasma treatment. A significant increase in cell proliferation was observed in the plasma-treated group (Group PS) on day 3 compared to the non-treated group (Group NS), indicating enhanced cytocompatibility.

This result supports that the plasma-treated SLA surface did not induce cytotoxicity and was favorable for cell growth, thereby suggesting the safety of the surface modification before applying it in vivo. and In this study, we focused primarily on evaluating local bone regeneration and osseointegration at the implantation sites. Cytotoxicity tests and systemic toxicity evaluations, including histopathological assessments of major organs such as the brain, heart, liver, and kidneys, were not performed as part of the current experimental design. However, no clinical signs of systemic adverse effects or abnormal behavior were observed in any of the animals during the study period. We acknowledge the importance of evaluating systemic biocompatibility and toxicity, and future studies will be designed to include comprehensive toxicological assessments.

  1. After the in vivo tests, did the authors check the implant’s morphologies and mechanical properties such as strengths and fatigues?

Response: Thank you for your insightful comment. In this study, the primary focus was to evaluate the biological response of implants—specifically bone regeneration and osseointegration—using imaging and histological analyses. Therefore, post-explant evaluation of implant surface morphology and mechanical properties such as strength and fatigue resistance was not conducted. We agree that such assessments could provide valuable information regarding the integrity and durability of the implants following in vivo use, and we will consider including these analyses in future studies.

  1. Scales in Figure 5 are too small and not clear. Figure 10 is not clear, please assure the image quality with at least 300 dpi. Figure 11 did not provide the scales.

Response: Thank you for your valuable comments. We have addressed each of the issues as follows:

Figure 5: The scale bars have been revised to ensure they are clearly visible and appropriately sized in the figure.

Figure 10: As this figure is based on medical CT images, the original resolution is inherently limited due to the nature of the imaging modality. Despite attempts, significant enhancement of image clarity was not achievable.

Figure 11: Scale bars have now been added to each SEM image in Figure 11 to clarify the magnification levels and provide spatial reference.

We hope these modifications adequately address your concerns.

  1. Is composition % in Table 1 weight or molecule? It is not clear.

Response: We appreciate the reviewer’s comment. According to the suggestion, we have clarified the unit in Table 1 by specifying that the composition percentages are presented as atomic percent (at%). The table has been revised accordingly to avoid confusion.

It is only my own opinion that the current manuscript cannot be considered for a publication in this journal before detailed addressing all the raised comments.

Response: We truly appreciate the reviewers’ thoughtful suggestions, which have helped us to significantly improve the quality and clarity of our manuscript. We hope the revised version satisfactorily addresses all the concerns raised.

Round 2

Reviewer 1 Report

Comments and Suggestions for Authors

The authors have corrected the errors/typos and accordingly revised the manuscript or provided sufficient explanations. 

Reviewer 2 Report

Comments and Suggestions for Authors

The authors have clearly addressed all the raised comments. Therefore, I don’t have any further scientific queries.

Comments on the Quality of English Language

The authors have clearly addressed all the raised comments. Therefore, I don’t have any further scientific queries.